# Clinical Factors Associated with Reinfection versus Relapse in Infective Endocarditis: Prospective Cohort Study

**DOI:** 10.3390/jcm10040748

**Published:** 2021-02-13

**Authors:** Jorge Calderón-Parra, Martha Kestler, Antonio Ramos-Martínez, Emilio Bouza, Maricela Valerio, Arístides de Alarcón, Rafael Luque, Miguel Ángel Goenaga, Tomás Echeverría, Mª Carmen Fariñas, Juan M. Pericàs, Guillermo Ojeda-Burgos, Ana Fernández-Cruz, Antonio Plata, David Vinuesa, Patricia Muñoz

**Affiliations:** 1Unidad de Enfermedades Infecciosas, Hospital Universitario Puerta de Hierro- Majadahonda (IDIPHSA), 28222 Madrid, Spain; jorge050390@gmail.com (J.C.-P.); anafcruz999@gmail.com (A.F.-C.); 2Servicio de Microbiología Clínica y Enfermedades Infecciosas, Hospital General Universitario Gregorio Marañón, 28007 Madrid, Spain; kestler.martha@gmail.com (M.K.); emilio.bouza@gmail.com (E.B.); mavami_valerio@yahoo.com.mx (M.V.); pmuñoz@micro.hggm.es (P.M.); 3Enfermedades Respiratorias-CIBERES (CB06/06/0058), Facultad de Medicina, Universidad Complutense de Madrid, 28040 Madrid, Spain; 4Clinical Unit of Infectious Diseases, Microbiology and Preventive Medicine Infectious Diseases Research Group, University of Seville/CSIC/University Virgen del Rocío and Virgen Macarena (IBIS), 41013 Sevilla, Spain; aa2406ge@yahoo.es (A.d.A.); rafaeluquemarquez@gmail.com (R.L.); 5Servicio de Enfermedades Infecciosas, Hospital Universitario Donostia, 20010 San Sebastián, Spain; goenagasanchez@gmail.com; 6Servicio de Cardiología, Hospital Donosti, 20010 San Sebastián, Spain; tomas.echeverriagarcia@osakidetza.eus; 7Infectious Diseases Unit, Hospital Universitario Marqués de Valdecilla, University of Cantabria, 39008 Santander, Spain; mcfarinas@humv.es; 8Infectious Disease Department, Hospital Clínic de Barcelona (IDIBAPS), 08036 Barcelona, Spain; pericasjm@gmail.com; 9Unidad de Gestión Clínica de Enfermedades Infecciosas, Hospital Universitario Virgen de la Victoria, 29010 Málaga, Spain; guilleojeda@gmail.com; 10Servicio de Enfermedades Infecciosas, Hospital Regional de Málaga, 29010 Málaga, Spain; antonio-plata@hotmail.com; 11Servicio de Medicina Interna y Enfermedades Infecciosas, Hospital Clínico San Cecilio, 18016 Granada, Spain; vinudav@yahoo.es

**Keywords:** endocarditis, bacterial, recurrence, *Enterococcus*, bacteremia, cardiac surgical procedures, liver disease

## Abstract

We aimed to identify clinical factors associated with recurrent infective endocarditis (IE) episodes. The clinical characteristics of 2816 consecutive patients with definite IE (January 2008–2018) were compared according to the development of a second episode of IE. A total of 2152 out of 2282 (94.3%) patients, who were discharged alive and followed-up for at least the first year, presented a single episode of IE, whereas 130 patients (5.7%) presented a recurrence; 70 cases (53.8%) were due to other microorganisms (reinfection), and 60 cases (46.2%) were due to the same microorganism causing the first episode. Thirty-eight patients (29.2%), whose recurrence was due to the same microorganism, were diagnosed during the first 6 months of follow-up and were considered relapses. Relapses were associated with nosocomial endocarditis (OR: 2.67 (95% CI: 1.37–5.29)), enterococci (OR: 3.01 (95% CI: 1.51–6.01)), persistent bacteremia (OR: 2.37 (95% CI: 1.05–5.36)), and surgical treatment (OR: 0.23 (0.1–0.53)). On the other hand, episodes of reinfection were more common in patients with chronic liver disease (OR: 3.1 (95% CI: 1.65–5.83)) and prosthetic endocarditis (OR: 1.71 (95% CI: 1.04–2.82)). The clinical factors associated with reinfection and relapse in patients with IE appear to be different. A better understanding of these factors would allow the development of more effective therapeutic strategies.

## 1. Introduction

Recurrent infective endocarditis (IE) is a feared complication that is associated with increased mortality [1,2]. Despite current therapies and prophylactic measures to prevent further episodes of IE, about 5–10% of patients eventually develop this condition [1,3,4,5].

Recurrent episodes of IE are classified as relapse or reinfection depending on the etiology and pathogenesis. The lack of eradication of the infection in the valve or adjacent tissue may be the cause of relapse of IE [6]. Conversely, reinfection is associated with new episodes of bacteremia in patients, where a condition predisposing to the development of IE may also be present [7]. Episodes of IE recurrences caused by a different microorganism than the previous one are usually considered as reinfections. On the other hand, when the new episode is caused by the same species, there is an inclination to consider it a relapse. According to molecular biology studies, the probability of the same strain causing a relapse is greater if recurrent IE occurs during the first six months after the initial episode [8]. Beyond that period there is uncertainty about its pathogenesis.

To date, most studies investigating recurrent episodes of endocarditis have not differentiated between relapse and reinfection, thereby preventing the identification of risk factors associated with each of them [1,2,3,5,9,10]. A recent study that specifically analyzed the risk factors of these two types of infections included a high number of intravenous drug users (IDUs), a rare practice in Spain in recent years [4]. Our aim was to identify the clinical factors associated with the development of relapse or reinfection in a large prospective and recent nationwide cohort of IE.

## 2. Methods

### 2.1. Design and Setting

A prospective cohort study, including cases of definite IE according to modified Duke criteria, was performed from January 2008 to June 2018 in the Games Cohort (Supportive Group for the Management of Infective Endocarditis in Spain). This registry was formed and maintained by 27, mainly tertiary, Spanish hospitals. Multidisciplinary “Endocarditis teams” in each participating institution completed standardized case report forms for subjects presenting IE episodes and follow-up data that included clinical, microbiological, and echocardiographic sections [11]. The database was continuously reviewed by a coordinator and a data manager who were responsible for contacting the different hospitals to keep the information accurately updated. Regional and local ethics committees approved the study, and all patients gave their informed consent.

### 2.2. Definitions

IE was defined using the modified Duke criteria [12]. Microbiological diagnosis was determined by blood or valve culture [11]. Transthoracic (TTE) and transesophageal (TEE) echocardiography were performed on patients with clinical or microbiological suspicion of IE according to European guidelines, or to diagnose valve dysfunction and intracardiac complications, such as abscesses, vegetation, pseudoaneurysms, or fistulae [13]. To consider hospital-acquired, non-healthcare-related, and community-acquired IE, definitions from previous studies were followed [10]. All necessary variables were collected to calculate the Charlson Comorbidity Index [14]. Recurrent episodes of IE were classified as relapses or reinfections. Relapses were defined as episodes of recurrent IE caused by the same organism detected within the first six months after completing the initial treatment and reinfections as new episodes of IE caused by a different microorganism within the follow-up period [13,15]. Although it is generally considered that repeated episodes of IE caused by the same species, but that appear after the first six months of follow-up, represent reinfections, a study based on molecular biology showed that some of these cases may correspond to relapses [8,13]. Due to this lack of certainty, it was decided not to include in either of the two groups the cases caused by the same species after the first six months of follow-up. No molecular biology studies were performed to corroborate that the cases considered as relapses were caused by the same strain. Only the first episode of recurrent endocarditis was analyzed. Exclusion criteria were possible IE according to modified Duke criteria, death during first IE episode admission, and less than 12 months’ follow-up after discharge.

The Cockcroft–Gault equation was used to calculate creatinine clearance [16]. Pre-episode renal insufficiency was defined as plasma creatinine over 1.4 mg/dL. New or worsening renal insufficiency during an IE episode was defined as a decrease in baseline creatinine clearance, or a minimum 25% increase in plasma creatinine, or creatinine levels over 1.4 mg/dL, when a previous analysis had been normal. Persistent bacteremia was defined as positive blood cultures more than seven days after effective antibiotic therapy.

### 2.3. Statistical Analysis

Quantitative variables were reported as median and interquartile ranges (IQRs) and qualitative variables as figures and percentages. In the comparisons with the different types of patients, only the variables corresponding to the first episode were analyzed. Continuous variables were compared with Student’s *t*-test, and categorical variables were compared using the chi-square test or Fisher’s exact test when appropriate. Stepwise logistic regression analyses were performed including variables present at admission with a *p* value < 0.1 in the univariate analysis, but also taking into account the clinical significance of each variable and the number of patients that reported the studied event. A two-sided *p* < 0.05 was considered to be statistically significant. Statistical analyses were performed using SPSS version 18 software (SPSS Inc., Chicago, IL, USA).

## 3. Results

During the study period, 2816 consecutive patients with definite IE were identified. During admission, 265 (9.4%) died, and 269 (9.6%) were lost or did not complete 12 months of follow-up. Out of the remaining 2282 subjects, 2152 (94.3%) had a single IE episode and 130 (5.7%) a second episode (Figure 1). The mean follow-up of IE patients was 3.4 years (range, 1–6.25 years).

The annual risk of suffering from a recurrent episode of IE in our study was 0.62% per patient-year. In the group that suffered a second episode, the causative microorganism was the same as in the initial episode in 60 (2.6%), and a different microorganism in 70 (3.1%). The clinical characteristics of the initial episode of IE, depending on whether the recurrence was caused by the same or different species, are shown in the Appendix A. Of the 60 patients who had a new episode of IE caused by the same species, 38 (1.7%) were diagnosed during the first six months of follow-up (and were considered to definitely suffer a relapse) and 22 after the first six months of follow-up (0.9%). The proportion of recurrent IE episodes caused by the same species during the first six months after completing treatment was 63.3% (38 out of 60 cases, *p* = 0.005), 36.4% between the 7th and 12th months (8 out of 22 cases, *p* = 0.437), and 29.1% when the episode appeared after the first year (14 out of 48 (29.1%), *p* = 0.005, Figure 2). Fifty-six patients (53.1%) underwent surgery after being diagnosed with recurrent IE. Subjects with liver cirrhosis, parenteral drug users, cases of prosthetic endocarditis, and patients who did not undergo surgical treatment had an increased risk of a second IE episode (Table 1). When the same variables were used for multivariate analyses, only liver cirrhosis and prosthetic endocarditis were associated with recurrent episodes (Table 1).

The characteristics of patients who suffered relapses (due to the same microorganism during the first six months after treatment) during the initial IE episode were compared with those who had a single episode of IE (Table 2). Individuals who presented with a relapse of the initial IE showed differences in a series of variables, such as nosocomial acquisition of the infection, IE due to *Enterococcus* spp., persistent bacteremia, and not receiving surgical treatment. Among the 14 patients who had a relapse of IE and who had surgical indications, 7 patients (50%) did not undergo surgery. Reasons for not carrying out cardiac surgery when indicated included liver cirrhosis (three cases), technical complexity (two cases), and hemodynamic instability and patient rejection (one case each). None of the patients whose relapse was due to enterococci underwent surgery, compared to 49% of those who had only one episode (*p* < 0.001). The following variables were included in the multivariable analysis, which showed that all were independently associated with the relapses: hospital-acquired IE, enterococcal IE persistent bacteremia, and not performing treatment when indicated (Table 2).

Compared to patients with a single IE episode, patients suffering reinfection had significantly higher rates of liver cirrhosis, prosthetic IE, and IE caused by anaerobic bacteria, whereas they presented significantly lower rates of IE caused by *S. aureus* (Table 2). The variables included in the multivariable analysis were liver cirrhosis, prosthetic endocarditis, and IE caused by anaerobic bacteria. Finally, all of them were significant variables (Table 2).

The relationship between the microbiology of the first and second episodes is shown in a Appendix A.

## 4. Discussion

This study represents the largest cohort of patients with IE and examines the clinical profile of recurrent endocarditis. The large number of subjects in our national cohort allowed us to identify clinical factors associated with recurrent IE in general, and for relapses and reinfections separately.

### 4.1. Clinical Factors Associated with Recurrence (Reinfection or Relapse)

We found a recurrence rate of 6%. This figure is similar to those of other series, with recurrent endocarditis ranging between 4% and 16% [2,3,5,10,17]. The proportion of relapses was lower than reinfections (1.7% vs. 3.1%), as observed in most analyses, particularly in those with longer follow-up periods [1,2,3,5,17]. Chronic liver disease and prosthetic endocarditis were associated with recurrent IE. Notably, we did not find that age was associated with recurrences, as prior studies did [5,18].

### 4.2. Clinical Factors Associated with Relapse

Relapse episodes suggest that treatment of the initial episode has been unsuccessful because of insufficient antimicrobial or surgical treatment that may allow a focus of infection (cardiac or metastatic) to remain. In our cohort, healthcare-associated IE, enterococcal etiology, persistent bacteremia, and not receiving cardiac surgery were risk factors for relapse. Alagna et al. also found that healthcare acquisition was associated with relapses [3]. It is likely that healthcare acquisition, enterococcal etiology, and low rates of cardiac surgery are tightly intertwined factors occurring mostly in elderly and fragile patients with high surgical risk [13]. Moreover, the significantly higher rates of relapse in enterococcal IE are an increasingly known phenomenon [19]. The observation that none of the patients who presented a relapsing enterococcal IE had undergone surgery reinforces the relevant role of surgical intervention in preventing relapses of enterococcal IE [20]. Further research is needed to elucidate the role of the type and length of antibiotic treatment in the frequency of relapses in enterococcal IE [19,21]. On the other hand, the relationship between persistent bacteremia and relapses is not surprising. The delay in microbiological eradication may be related to the initial high level of bacterial inoculum in the bloodstream, an uncontrolled focus of infection, the use of an antibiotic regimen that is not fully effective (due to inadequate penetration of the drug into vegetation or difficult-to-treat microorganisms), or the development of resistance during treatment. All these circumstances could delay the definitive cure of the infection and favor the development of a relapse [1].

### 4.3. Clinical Factors Associated with Reinfection

Reinfection is related to the occurrence of repetitive episodes of bacteremia in patients who may have heart or vascular conditions that predispose them to implantation and growth of bacteria. Reinfection was significantly associated with prosthetic IE, chronic liver disease, and IE due to anaerobic bacteria. The increased risk of recurring IE in patients with prosthetic valves had already been established in prior studies [2,9,12], and it is likely linked to the ability of bacteria to adhere to artificial surfaces, such as the sewing ring, and the tissue damage caused by prior IE episodes [3,9].

It has been shown that patients suffering from liver cirrhosis are at higher risk of developing IE than the general population. Similarly, the percentage of patients intervened is low and mortality high (particularly Child stage C subjects) [22]. The increased risk of reinfection in these patients may be related to increased gut permeability, predisposition to bacteremia, immune dysfunction, and a frequent need for invasive procedures [23]. Conducting randomized studies to clarify the usefulness of antibiotic prophylaxis in endoscopic procedures, such as esophageal variceal ligation, seems to be an interesting goal, but it has to cope with the reduced number of cases of infectious endocarditis in cirrhotic patients [22,24].

To date, there have been no studies that have analyzed the possible tendency to experience reinfection of IE due to anaerobes. These patients more often are men with poor dental hygiene and a prosthetic valve [25]. These cases frequently present large vegetation with extensive valvular destruction and congestive heart failure [26]. It is possible that the persistence of an unresolved polymicrobial and oligosymptomatic septic focus located in the digestive tract could be related to this type of reinfection. In any case, the reduced number of patients with this complication puts constraints on the importance of this finding.

Some variables related to reinfection observed in other studies, such as intravenous addiction or hemodialysis, may not have been detected because of the reduced number of patients who were exposed to these risk factors in our study [3,4].

### 4.4. Relationship between the Microbiology of the First Episode and That of the Second Episode in Patients with Reinfection

An innovative aspect of our study was the relationship between the microbiology of the first and second episodes in reinfected patients. The potential effect of antibiotics on the etiology of subsequent IE episodes has been observed previously [27]. Our results would suggest that previous treatment with cephalosporins (in the context of streptococcal endocarditis) may have favored the etiology of a second episode of IE to be enterococcal and that treatment for enterococci may have promoted the appearance of staphylococci in the next episode of IE.

### 4.5. Limitations

There are limitations to this study that need consideration. The main limitation is that no molecular biology studies were performed to corroborate that the cases considered as relapses were caused by the same strain. Another limitation is that patients with a new episode of IE caused by the initial bacterial species, where detection occurred six months after treatment, could not be reliably assigned to either group (relapses or reinfections). We would also like to draw attention to the low number of studied patients (especially in the relapse group) that may have prevented the identification of other associated clinical factors.

## 5. Conclusions

A better understanding of the clinical factors associated with recurrent episodes of IE can be helpful in informing patients and providers of the risks of reinfection and relapse. Although some of these factors are not modifiable, possible areas of research are identified, such as the role of surgery in cases where there is no established indication for surgery, but a high risk of relapse is deemed. Another line of future research is to study the best preventive strategy with regard to reinfection in patients with chronic liver disease or a history of prosthetic endocarditis.

## Figures and Tables

**Figure 1 jcm-10-00748-f001:**
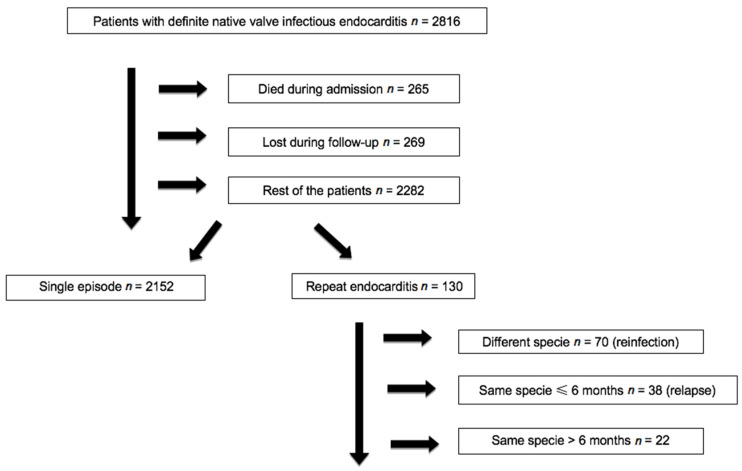
Flowchart showing patients presenting with definite valve infective endocarditis according to suffering from a repeat episode (Games cohort).

**Figure 2 jcm-10-00748-f002:**
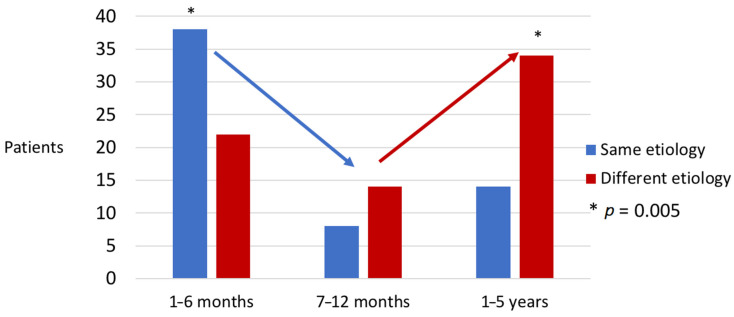
Distribution of cases according to etiology of recurrent IE and the time elapsed since the first episode. * *p* = 0.005.

**Table 1 jcm-10-00748-t001:** Characteristics of the initial episode in patients who developed recurrent infective endocarditis in comparison with patients who presented one episode.

	Recurrent Endocarditis (*n* = 130)	One Episode (*n* = 2152)	OR (95% CI) ^1^	*p*
Age (years)	65 (46–74)	66 (53–75)		0.222
Male gender	88 (67.7)	1505 (69.9)		0.658
Hospital-acquired	41 (31.5)	516 (23.9)		0.065
Non-nosocomial healthcare related	14 (10.7)	175 (8.1)		0.370
Community-acquired	75 (57.7)	1461 (67.9)		0.021
Diabetes mellitus	32 (24.6)	538 (25.0)		0.995
Coronary disease	37 (28.5)	541 (25.2)		0.458
Peripheral arterial disease	9 (6.9)	196 (9.1)		0.491
Cerebrovascular disease	22 (16.9)	240 (11.1)		0.062
Previous renal failure	23 (17.7)	410 (19.1)		0.701
Chronic hemodialysis	5 (3.8)	79 (3.6)		0.891
Chronic liver disease	21 (16.3)	155 (7.2)	2.34 (1.39–3.9)	<0.01
Injection drug user	8 (6.2)	47 (2.2)	2.06 (0.87–4.93)	0.004
Neoplasia	22 (16.9)	281 (13.0)		0.259
Age-adjusted Charlson Comorbidity Index (points)	4 (2–6)	4 (2–6)		0.819
Site of infection				
Native valve	67 (51.5)	1317 (61.2)		0.029
Prosthetic valve	47 (36.2)	580 (27.0)	1.64 (1.12–2.39)	0.022
Cardiac device	15 (11.5)	277 (12.9)		0.659
Involved valve				
Mitral	69 (53.1)	1041 (48.4)		0.297
Aortic	52 (40.0)	862 (40.1)		0.990
Tricuspid	6 (4.6)	131 (6.1)		0.493
Pulmonary	3 (2.3)	26 (1.2)		0.277
Microbiology				
Gram-positive bacteria				
Coagulase-negative staphylococci	22 (16.9)	332 (15.4)		0.647
*S. aureus*	22 (16.9)	415 (19.3)		0.506
*Enterococcus* spp.	25 (19.2)	302 (14.0)		0.101
*Streptococcus* spp.	30 (23.1)	654 (30.4)		0.077
Gram-negative bacilli	9 (6.9)	89 (4.1)		0.128
Anaerobic bacteria	4 (3.1)	28 (1.3)		0.094
*Candida*	1 (0.8)	23 (1.1)		0.745
Polymicrobial	1 (0.8)	38 (1.8)		0.395
Other microorganisms	18 (13.8)	224 (10.4)		0.216
Negative cultures	11 (8.5)	186 (8.6)		0.943
Septic shock	5 (3.8)	129 (5.9)		0.412
Persistent bacteremia	15 (11.5)	206 (9.5)		0.559
CNS vascular events	19 (14.6)	318 (14.7)		0.938
Embolism	28 (21.5)	435 (20.2)		0.800
Heart failure	43 (33.0)	651 (30.2)		0.560
New or worsening renal insufficiency	39 (30.0)	588 (27.3)		0.573
Echocardiographic findings				
Vegetation	86 (66.2)	1532 (71.2)		0.216
Perivalvular abscess	17 (13.2)	272 (12.7)		0.970
Valve perforation or rupture	14 (10.7)	281 (13.2)		0.448
Pseudoaneurysm	4 (3.1)	101 (4.7)		0.692
Intracardiac fistula	3 (2.3)	43 (2.0)		0.966
Surgical indication	75 (57.6)	1319 (61.2)		0.468
Surgery performed	56 (43.1)	1119 (52.0)	0.74 (0.52–1.1)	0.048
Surgery indicated but not performed	19 (14.6)	200 (9.3)		0.232
Device extraction	9 (60)	233 (84.1)		0.016
Duration of antibiotic treatment	42 (32–50)	42 (30–47)		0.323
Hospital stay (days)	40 (27–53)	40 (25–54)		0.766

CNS: central nervous system. Quantitative variables are reported with median and interquartile range. ^1^ Multivariate analysis.

**Table 2 jcm-10-00748-t002:** Characteristics of the initial episode in patients who developed recurrences in comparison with patients who presented one episode.

	One Episode (*n* = 2152)	OR (95% CI) ^1^	*p*	Relapses (*n* = 38)	OR (95% CI) ^1^	*p* ^2^	Reinfection (*n* = 70)
Age (years)	66 (53–75)		0.554	67 (59–77)		0.066	64 (45–73)
Male gender	1505 (69.9)		0.469	24 (63.1)		0.712	47 (67.1)
Hospital-acquired	516 (23.9)	2.67 (1.37–5.29)	0.001	18 (47.3)		0.531	14 (20.0)
Non-nosocomial healthcare related	175 (8.1)		0.805	3 (7.8)		0.233	9 (12.8)
Community-acquired	1461 (67.9)		<0.001	17 (44.7)		0.986	47 (67.1)
Diabetes mellitus	538 (25.0)		0.997	10 (26.3)		0.416	14 (20.0)
Coronary disease	541 (25.2)		0.728	11 (28.9)		0.609	20 (28.6)
Peripheral arterial disease	196 (9.1)		0.272	1 (2.6)		0.724	5 (7.1)
Cerebrovascular disease	240 (11.1)		0.252	7 (18.4)		0.904	8 (11.4)
Previous renal failure	410 (19.1)		0.477	5 (13.2)		0.584	11 (15.7)
Chronic hemodialysis	79 (3.6)		0.444	0		0.957	3 (4.2)
Chronic liver disease	155 (7.2)		0.090	6 (15.8)	3.1 (1.65–5.83)	0.009	13 (18.6)
Injection drug user	47 (2.2)		0.721	0		0.124	4 (5.7)
Neoplasia	281 (13.0)		0.239	2 (5.3)		0.064	15 (21.4)
Age-adjusted Charlson Comorbidity Index (points)	4 (2–6)		0.313	5 (2–6)		0.318	4 (1–5)
Site of infection							
Native valve	1317 (61.2)		0.216	19 (50.0)		0.199	37 (52.9)
Prosthetic valve	580 (27.0)		0.124	15 (39.5)	1.71 (1.04–2.82)	0.044	27 (38,6)
Cardiac device	277 (12.9)		0.854	4 (10.5)		0.378	6 (8,6)
Involved valve							
Mitral	1041 (48.4)		0.721	20 (52.6)		0.276	39 (55.7)
Aortic	862 (40.1)		0.929	16 (42.1)		0.557	31 (44.3)
Tricuspid	131 (6.1)		0.904	3 (7.9)		0.386	2 (2.9)
Pulmonary	26 (1.2)		0.962	1 (2.6)		0.697	1 (1.4)
Microbiology							
Gram-positive bacteria							
Coagulase-negative staphylococci	332 (15.4)		0.293	3 (7.9)		0.383	14 (20.0)
*S. aureus*	415 (19.3)		0.198	11 (28.9)		0.036	6 (8.6)
*Enterococcus* spp.	302 (14.0)	3.01 (1.51–6.01)	<0.001	14 (36.8)		0.657	8 (11.4)
*Streptococcus* spp.	654 (30.4)		0.077	6 (15.8)		0.652	19 (27.1)
Gram-negative bacilli	89 (4.1)		0.125	4 (10.5)		0.353	5 (7.1)
Anaerobic bacteria	28 (1.3)		0.983	0	4.12 (1.4–12.4)	0.011	4 (5.7)
*Candida* spp.	23 (1.1)		0.871	0		0.763	1 (1.4)
Polymicrobial	38 (1.8)		0.408	0		0.801	1 (1.4)
Other microorganisms	224 (10.4)		0.813	3 (7.9)		0.047	13 (18.6)
Negative cultures	186 (8.6)		0.109	0		0.311	9 (12.9)
Septic shock	129 (5.9)		0.600	1 (2.6)		0.735	3 (4.3)
Persistent bacteremia	206 (9.5)	2.37 (1.05–5.36)	0.036	8 (21.1)		0.941	6 (8.6)
CNS vascular events	318 (14.7)		0.615	4 (10.5)		0.706	12 (17.1)
Embolism	435 (20.2)		0.128	12 (31.6)		0.439	11 (15.7)
Heart failure	651 (30.2)		0.996	11 (28.9)		0.062	29 (41.4)
New or worsening renal insufficiency	588 (27.3)		0.066	16 (42.1)		0.918	19 (27.1)
Echocardiographic findings							
Vegetation	1532 (71.2)		0.985	27 (71.1)		0.934	49 (70.0)
Perivalvular abscess	272 (12.7)		0.530	3 (7.9)		0.353	12 (17.4)
Valve perforation or rupture	281 (13.2)		0.822	5 (13.2)		0.827	8 (11.6)
Pseudoaneurysm	101 (4.7)		0.834	1 (2.6)		0.897	3 (4.3)
Intracardiac fistula	43 (2.0)		0.771	0		0.370	3 (4.3)
Surgical indication	1319 (61.2)		0.003	14 (36.8)		0.111	50 (71.4)
Surgery performed	1119 (52.0)	0.23 (0.1–0.53)	0.001	7 (18.4)		0.231	42 (60.0)
Surgery indicated not performed	200 (9.3)		0.211	7 (18.4)	1.03 (0.49–2.21)	0.971	8 (11.4)
Duration of antibiotic treatment	40 (25–54)		0.662	40 (27–56)		0.655	42 (28–53)
Hospital stay (days)	42 (30–47)		0.882	42 (30–49)		0.092	42 (37–49)

CNS: central nervous system. PVIE: prosthetic valve infectious endocarditis. Quantitative variables are reported with median and interquartile range. ^1^ Multivariate analysis. ^2^ Comparison of reinfection cases compared to cases that presented a single episode of infective endocarditis.

## Data Availability

The information contained in the database may be accessible after contacting the corresponding author and after its reasoned justification.

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
