# Peer review of "Clinical Factors Associated with Reinfection versus Relapse in Infective Endocarditis: Prospective Cohort Study"

_jcm, 2021, doi:10.3390/jcm10040748_

Round 1

Reviewer 1 Report

This interesting  article deals with a very important issue of IE. The research was planned prospectively and carried out on a relatively large group of patients. The results were presented in a clear and understandable way, I also have no objections to the discussion conducted. Certainly due to the importance of the topic and the limitations presented by the authors, the work needs to be continued. It seems interesting to supplement what kinds of cardiac devices were the cause  for relapse or reinfection. As it is well known, the frequency of IE is different in the presence of LVAD and PM.

Author Response

We appreciate the words of the reviewer. It is true that finding out the risk of recurrence according to the type of cardiac device is a very interesting question. Unfortunately, we do not have this information in our database. However, I am sure that most of the devices correspond to pacemakers or implantable cardioverter-defibrillator. In future studies on this subject, we must take this issue into account.

Reviewer 2 Report

The manuscript deals with very interesting and clinicaly important theme- recurrent infective endocarditis and clinical factors associated with recurrence in a large prospective and  nationwide cohort of IE.

It should be noted, that the study analysed the risk factors of the recurrence of IE  not in general, but distinguished two types of recurrent IE –reinfection and relapse, as most of earlier studies have not differentiated these two types of infection.

The aim of the study is presented clearly, the methods used, including design, definitions and statistical analysis are described thoroughly.

The results are also presented thoroughly, including 2 figures,2 huge tables and 2 tables in the supplement.

The discussion is rather short but concrete. However, a few items in the discussion- the association between recurrent IE and chronic liver disease could be expanded. The assumptions of the authors on  associations between all determined clinical factors,relapse and reinfection would be valuable.

The clinical factors associated with relapse are nicely discussed, moreover, the authors innovatively analyse the relationship between the microbiology of the first and second episodes in reinfection.

The conclusions are concrete and gives subtle suggestions for future research.

The English level of the manuscript is high. The manuscript will be interesting to the wide auditorium, especially to cardiologists and cardiac surgeons and microbiologists.

Author Response

Some considerations have been added on the risk of reinfection in patients with liver cirrhosis and on some preventive EI strategies in these patients. A paragraph has been added to attempt to explain the increased risk of reinfection in cases due to anaerobic species. To explain recurrent episodes of IE due to reinfection, certain characteristics of patients with anaerobic infection in other parts of the body have been assumed

Reviewer 3 Report

This study report clinical and biological factors associated with the recurrence of endocarditis in patients who have recovered from a previous episode of endocarditis. The cohort of studied patients is important and results are thus sufficiently substantiated. The study is well constructed Results and discussion rely on three successive analyses: Factors associated with recurrence, relapse or reinfection. However the first analyze (first episode of IE and recurrence) does not seem essential and could be removed. This analyze includes 22 unclassified patients and the two independent prognosis factors identified ( chronic liver disease and prosthetic endocarditis) are highlighted in the third analyze ( first episode of IE and reinfection)

There is no major concern about this study

Minor concerns:

- The Euroscore, as collected data, is announced in definitions but not reported on different tables

- Line 163 : “Sixty nine patients underwent surgery”. According to table 1, it seems that this only concern56 patients.

- Line 181 : there appear to be an error: it seems that it is non surgery that is associated with reinfection

- Line 186 : “prosthetic endocarditis caused by anaerobic bacteria” as independent factors for reinfection. For me, these two factors are independent and not related to each other as implied by the sentence

- As persistence of bacteremia is associated with relapse, is it possible to have a definition with the number of days of positivity. It is the same problem for chronic liver disease: is it possible to better define these patients (liver cirrhosis ?)

- We expected, as in other studies, that S. aureus could be associated with recurrence. Is it possible to differentiate between methicillin-sensible S. aureus and methicillin-resistant S. aureus ?

- Line 254 : ref 25 and 26 are announced, but there is no ref 26 in references

Author Response

This study report clinical and biological factors associated with the recurrence of endocarditis in patients who have recovered from a previous episode of endocarditis. The cohort of studied patients is important and results are thus sufficiently substantiated. The study is well constructed Results and discussion rely on three successive analyses: Factors associated with recurrence, relapse or reinfection. However the first analyze (first episode of IE and recurrence) does not seem essential and could be removed. This analyze includes 22 unclassified patients and the two independent prognosis factors identified ( chronic liver disease and prosthetic endocarditis) are highlighted in the third analyze ( first episode of IE and reinfection)

We agree with the reviewer that the specific analysis on patients with recurrences (all patients with a second episode, regardless of whether they are relapses or reinfections) is less interesting than later ones. However, as it is based on a larger number of patients (130) we believe that the results are more robust, and we advocate leaving it in the final version of the manuscript. We also consider that it provides information that facilitates comparison with previous studies that do not differentiate between relapses and reinfections. If it were regarded as an unsurmountable obstacle, we could move Table 1 to the supplementary material.

There is no major concern about this study

Minor concerns:

- The Euroscore, as collected data, is announced in definitions but not reported on different tables

With regard to this point, we thought it best to remove the references to the Euroscore from the article. It was not a variable influencing the risk of recurrent IE

- Line 163 : “Sixty nine patients underwent surgery”. According to table 1, it seems that this only concern56 patients.

Finally, the correct number is 56. Accordingly, this number has been corrected in the text.

- Line 181 : there appear to be an error: it seems that it is non surgery that is associated with reinfection

The phrase has been modified to clarify that it is actually the failure to perform surgery that is associated with relapses.

- Line 186 : “prosthetic endocarditis caused by anaerobic bacteria” as independent factors for reinfection. For me, these two factors are independent and not related to each other as implied by the sentence

That sentence is wrong, it should instead state: " The variables included in the multivariable analysis were chronic liver disease, prosthetic endocarditis and IE caused by anaerobic bacteria ". The phrase has been modified in this manner

- As persistence of bacteremia is associated with relapse, is it possible to have a definition with the number of days of positivity. It is the same problem for chronic liver disease: is it possible to better define these patients (liver cirrhosis ?)

In methods, we specify that persistent bacteremia was defined as positive blood cultures more than seven days after effective antibiotic therapy. We agree with the reviewer: "chronic liver disease" has been replaced by “liver cirrhosis” as a term more appropriate to the underlying diseases of our patients.

- We expected, as in other studies, that S. aureus could be associated with recurrence. Is it possible to differentiate between methicillin-sensible S. aureus and methicillin-resistant S. aureus ?

This was a surprising result from what was previously published. In our series S. aureus was associated with higher severity during the initial episode but not with higher risk of recurrences. We have no explanation for this result. Unfortunately, this data was not systematically collected in our database.

- Line 254 : ref 25 and 26 are announced, but there is no ref 26 in references

Reference # 26 has been removed from the text, although other references related to the comments that have been added to the discussion have been added.